

# The effects of an invasive seaweed on native communities vary along a gradient of land-based human impacts

Fabio Bulleri[1], Fabio Badalamenti[2], Ljiljana Iveša[3], Barbara Mikac[2], Luigi Musco[2], Andrej Jaklin[3], Alex Rattray[1], Tomás Vega Fernández[2,4] and Lisandro Benedetti-Cecchi[1]

[1] Department of Biology, University of Pisa, Pisa, Italy
[2] CNR-IAMC, Istituto per l'Ambiente Marino Costiero, Castellammare del Golfo, Italy
[3] Ruđer Bošković Institute, Center for Marine Research, G. Paliaga, Rovinj, Croatia
[4] Stazione Zoologica Anton Dohrn, Napoli, Italy

Corresponding author
Fabio Bulleri, fabio.bulleri@unipi.it

## ABSTRACT

The difficulty in teasing apart the effects of biological invasions from those of other anthropogenic perturbations has hampered our understanding of the mechanisms underpinning the global biodiversity crisis. The recent elaboration of global-scale maps of cumulative human impacts provides a unique opportunity to assess how the impact of invaders varies among areas exposed to different anthropogenic activities. A recent meta-analysis has shown that the effects of invasive seaweeds on native biota tend to be more negative in relatively pristine than in human-impacted environments. Here, we tested this hypothesis through the experimental removal of the invasive green seaweed, *Caulerpa cylindracea*, from rocky reefs across the Mediterranean Sea. More specifically, we assessed which out of land-based and sea-based cumulative impact scores was a better predictor of the direction and magnitude of the effects of this seaweed on extant and recovering native assemblages. Approximately 15 months after the start of the experiment, the removal of *C. cylindracea* from extant assemblages enhanced the cover of canopy-forming macroalgae at relatively pristine sites. This did not, however, result in major changes in total cover or species richness of native assemblages. Preventing *C. cylindracea* re-invasion of cleared plots at pristine sites promoted the recovery of canopy-forming and encrusting macroalgae and hampered that of algal turfs, ultimately resulting in increased species richness. These effects weakened progressively with increasing levels of land-based human impacts and, indeed, shifted in sign at the upper end of the gradient investigated. Thus, at sites exposed to intense disturbance from land-based human activities, the removal of *C. cylindracea* fostered the cover of algal turfs and decreased that of encrusting algae, with no net effect on species richness. Our results suggests that competition from *C. cylindracea* is an important determinant of benthic assemblage diversity in pristine environments, but less so in species-poor assemblages found at sites exposed to intense disturbance from land-based human activities, where either adverse physical factors or lack of propagules may constrain the number of potential native colonizers. Implementing measures to reduce the establishment and spread of *C. cylindracea* in areas little impacted by land-based human activities should be considered a priority for preserving the biodiversity of Mediterranean shallow rocky reefs.

## INTRODUCTION

Concerns over the potential of invasive species to alter biodiversity, impair ecosystem functioning and cause economic loss have stimulated research on the mechanisms underpinning variations in their impacts on native species and communities (*Mack et al., 2000*). Although a unified framework for predicting the direction and magnitude of invaders' impacts is still elusive, some generalities have started to emerge from qualitative and quantitative synthesis of the literature (*Schaffelke & Hewitt, 2007*; *Williams & Smith, 2007*; *Gaertner et al., 2009*; *Thomsen et al., 2009*; *Thomsen et al., 2014*; *Powell, Chase & Knight, 2011*; *Vilà et al., 2011*; *Maggi et al., 2015*; *Tamburello et al., 2015*). For instance, the effects of non-native plants on native plant communities have been found to be generally negative, whilst those on higher trophic-level communities or species can be less predictable, varying from negative to positive (*Vilà et al., 2011*; *Thomsen et al., 2014*; *Maggi et al., 2015*).

More recently, *Tamburello et al. (2015)* have shown, by means of a global meta-analysis, that the effects of non-native seaweeds on native benthic communities tend to shift from negative to neutral or positive along a gradient of increasing cumulative human impact (*Halpern et al., 2008*), suggesting that the severity of their effects might be greater in relatively pristine than in degraded environments. The impact score developed by *Halpern et al. (2008)* combines a diverse set of anthropogenic drivers, including climate change, land-based and sea-based human impacts. Nonetheless, it provides little insight into the nature of the key drivers underpinning variations in the effects of non-native seaweeds on native benthic assemblages. Land-based human impacts, such as inorganic pollution, enhanced nutrient loading and sedimentation have been widely shown to cause changes in the structure (i.e. species composition and relative abundance) of benthic assemblages on temperate rocky reefs (*Benedetti-Cecchi et al., 2001*; *Airoldi & Beck, 2007*; *Gorman, Russell & Connell, 2009*). Thus, it could be argued that land-based human impacts are more likely to influence the fitness and competitive ability of both native and non-native macroalgal species, as well as the outcome of their interactions, in comparison to sea-based impacts.

Here, we experimentally assessed how the effects of the invasive seaweed, *C. cylindracea* Sonder, vary among Mediterranean rocky reefs exposed to different levels of human impact. This seaweed, introduced in the NW Mediterranean about 20 years ago, has the potential to reduce the abundance of encrusting, turf-forming and erect macroalgae, ultimately decreasing species richness, total cover and spatial variability of native assemblages (*Piazzi, Ceccherelli & Cinelli, 2001*; *Bulleri et al., 2010*; *Gennaro & Piazzi, 2011*; *Papini, Mosti & Santosuosso, 2013*; *Tamburello et al., 2015*). Assessing the relative role of land-based (i.e. coastal human population, discharge of fertilizers and pesticides, river run-off and risk of hypoxia) versus sea-based (i.e. shipping activities, oil spill risk

and fishing activities) human impacts in regulating the direction and intensity of the effects of *C. cylindracea* on native assemblages is, however, key for prioritizing areas of management and invader control strategies.

We expected that the severity of land-based cumulative human impacts would be a better predictor than sea-based impacts of variations in the effects of *C. cylindracea* on the abundance of main morphological groups of benthic organisms and on the total cover and diversity of native assemblages. According to the findings of *Tamburello et al. (2015)*, we predicted the effects of *C. cylindracea* on native morphological group abundance and community structure to be negative in relatively pristine areas and become neutral or positive in areas characterized by higher scores of cumulative human impacts.

Due to the alteration of internal feed-back mechanisms, the changes caused by invasive species can be difficult to revert (*Gaertner et al., 2014*). For instance, enhanced sediment deposition in the presence of *C. cylindracea* (*Piazzi et al., 2007*) can lock the system into an "invaded" state characterized by the dominance of algal turfs (*Bulleri et al., 2010*). The removal of the invader and of the assemblages established under its dominance is, therefore, a crucial requirement of experimental tests of the impacts of biological invasions (*Bulleri et al., 2010*). For this reason, we also assessed the effects of *C. cylindracea* on the recovery of native assemblages in areas totally cleared at the beginning of the experiment.

## MATERIALS AND METHODS

This study was carried out at eight sites in the Mediterranean Sea (Fig. 1). Due to logistic constraints, study sites were not evenly distributed across the Mediterranean. Nonetheless, in order to encompass a wide gradient of human impacts, they were haphazardly chosen in the proximity of major urban or industrial centers, extra-urban sites, and along the coast of islands little exposed to human activities. At each site, twenty 40 × 40 cm plots were marked with epoxy putty (Veneziani S subcoat) at a depth of 5–8 m, in June 2012. Five plots were randomly assigned to each of the following four treatments generated by crossing disturbance (2 levels; control versus clearing of the whole assemblage, hereafter referred to as extant and recovering assemblages, respectively) and manipulation of *C. cylindracea* (2 levels; present versus removed): 1) removal of *C. cylindracea* from extant assemblages, 2) total clearing of extant assemblages preventing the re-invasion by *C. cylindracea*, 3) total clearing of extant assemblages but allowing re-invasion by *C. cylindracea* and 4) controls, assemblages invaded by *C. cylindracea*. Clearing of extant assemblages (treatments 2 and 3), consisting of the removal of erect organisms by means of a metal brush, was carried out once at the beginning of the study. Experimental conditions were maintained by manually removing *C. cylindracea* every 2–4 weeks. During visits in the field, the percentage cover of *C. cylindracea* was assessed in the central 20 × 20 cm area of each plot (to avoid edge effects), using a grid subdivided in 25 quadrats. A score from 0 to 4% was given to each sub-quadrat and the percentage cover was obtained by summing over the entire set of subquadrats. Although the natural cover of *C. cylindracea* varied among study sites, experimental

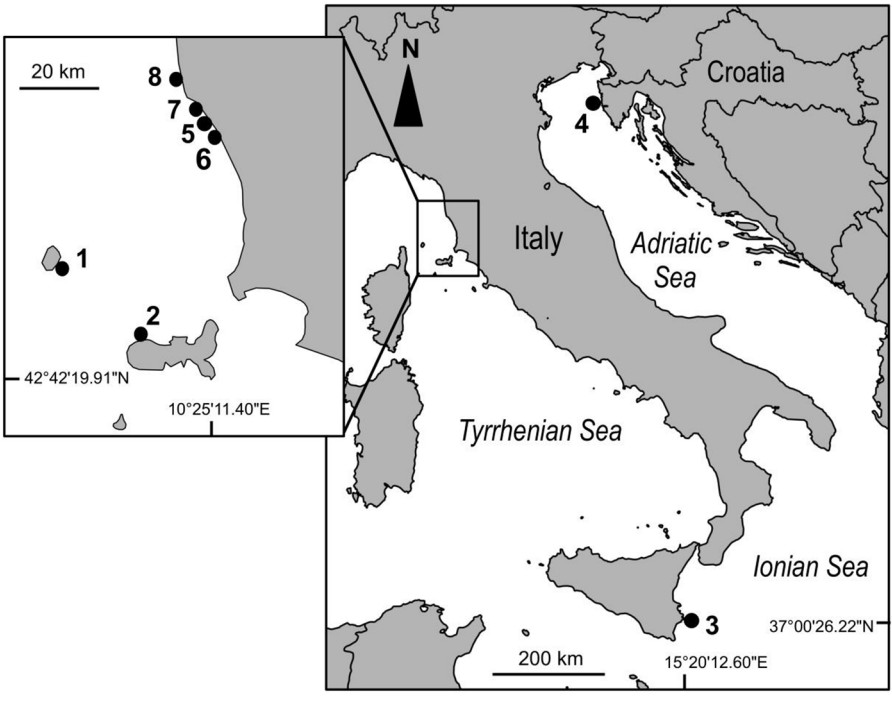

**Figure 1 Map of the study sites in the Mediterranean Sea.** The numbering of sites (1–8) follows an increasing gradient of land-based human impacts. 1: Capraia island (43°00′52.85″N; 9°49′27.52″E); 2: Elba island (42°48′33.18″N; 10°08′26.13″E); 3: Plemmirio (37°00′26.22″N; 15°20′12.6″E); 4: Vrsar (45°8′58.96″N; 13°35′28.38″E); 5: Fortullino (43°25′39.66″N; 10°23′41.04″E); 6: Rosignano Solvay (43°24′07.25″N; 10°24′47.44″E); 7: Quercianella (43°27′20.72″N; 10°22′04.48″E); 8: Livorno (43°31′18.90″N; 10°18′37.72″E).

conditions were maintained throughout the study period (Fig. 2). After ~15 months from the start of the experiment, the cover of macroalgae and sessile invertebrates was estimated by means of the same visual technique used to estimate that of *C. cylindracea*. Organisms were generally identified to the species level, except for encrusting coralline and filamentous algae. For analysis, macroalgal species were included into three major morphological groups: encrusting, turf-forming and canopy-forming (*Airoldi, Rindi & Cinelli, 1995*; *Benedetti-Cecchi et al., 2001*). Likewise, sessile invertebrates such as sponges, ascidians, bryozoans, hydrozoans, bivalves, tubiculous molluscs and polychaetes were analyzed as a single morphological group. With the exception of the filamentous macroalga, *Womersleyella setacea*, present in two plots at Site 4, species composing benthic assemblages were native. Fieldwork along the Italian coast was conducted with permits from Comune di Livorno (Prot. N° 71719), Ente Parco Arcipelago Toscano (Prot. N. 1797) and Area Marina Protetta Plemmirio (CNR-IAMC-Consorzio Plemmirio convention N° 0007139). Fieldwork along the Croatian coast was conducted with permits from the Ministry of Culture (UP/I-612-07/11-33/1015; 532-08-01-01/1-11-04; from 25 October 2011).

For each study site, using ArcGIS 10.1, we extracted a set of both land-based and sea-based anthropogenic drivers from the georeferenced layers developed by *Micheli et al. (2013)* for the Mediterranean basin. Land-based drivers included coastal population

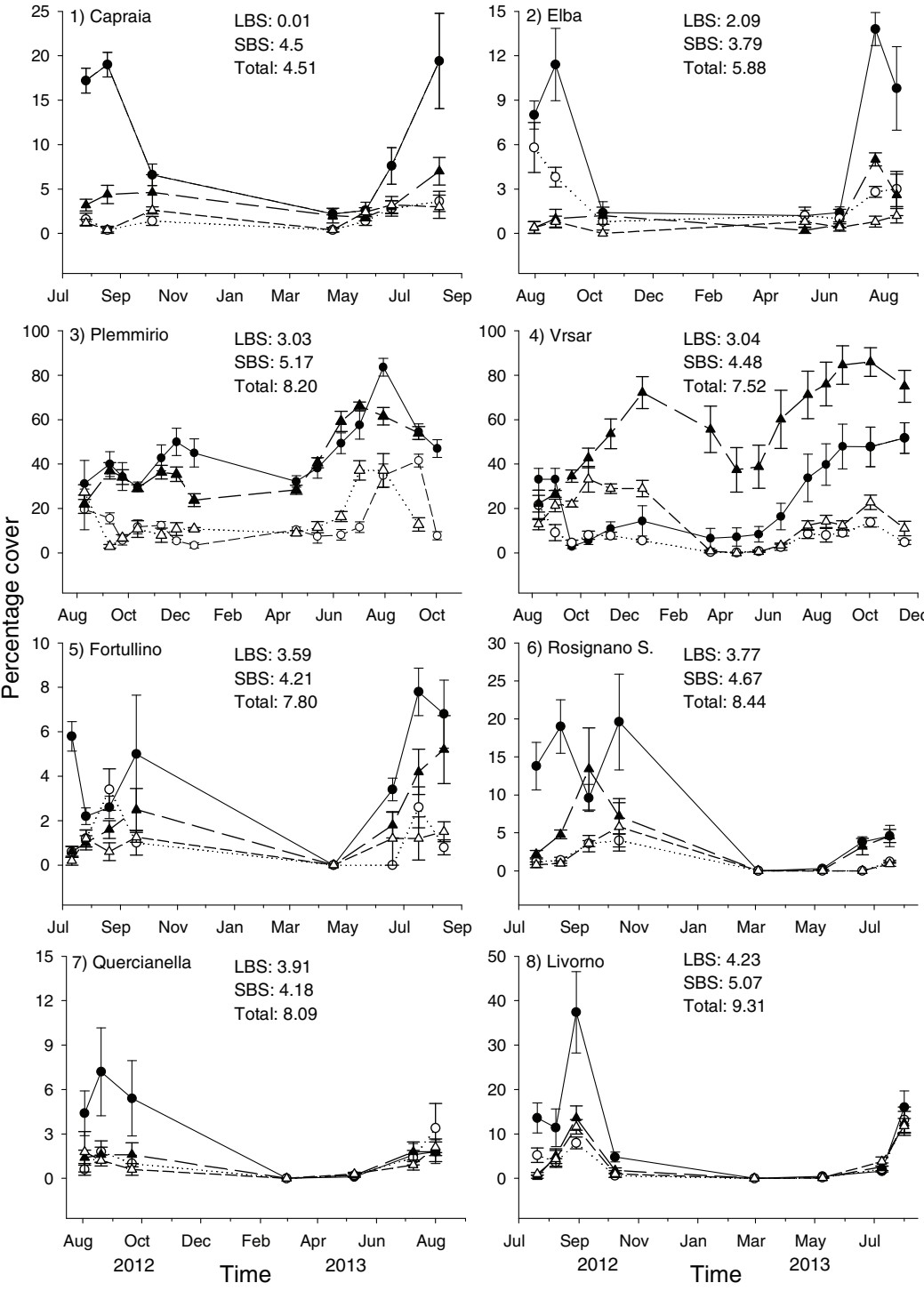

**Figure 2 Percentage cover of *C. cylindracea* in experimental plots at each site throughout the study.**
Filled circle = Extant assemblages + *C. cylindracea*; Hollow circle = Extant assemblages − *C. cylindracea*;
Filled triangle = Developing assemblages + *C. cylindracea*; Hollow triangle = Developing assemblages −
*C. cylindracea*. Cumulative Land-Based (LBS), Sea-Based (SBS) and total human impact scores are
reported for each study site.

density, nutrient input (fertilizers), organic pollution (pesticides), urban run-off and risk of hypoxia, whilst sea-based drivers included benthic structures (oil rigs), commercial shipping, invasive species, artisanal fishing, oil spills and 5 different types of commercial fishing (see Table S1 in *Micheli et al., 2013*). These drivers were included to take into account the potential human effects, either direct or indirect, on benthic habitats. Invasive species were included among the sea-based drivers since the importance of competitive effects of *C. cylindracea* on native assemblages can be hypothesized to be less severe in communities altered by the presence of multiple introduced species (*Tamburello et al., 2015*). Following *Micheli et al. (2013)*, we log (x + 1)-transformed and rescaled each driver layer between 0–1 to render them in a single, unitless scale that allows for direct comparison.

Both land- and sea-based cumulative impact scores at the northernmost (Vrsar) and southernmost (Plemmirio) sites were within the range found at study sites in the Northern Tyrrhenian and Ligurian Seas (i.e. they did not constitute extremes). When considering land-based human impacts, sites with higher scores were clumped along the mainland coast of Tuscany (Sites 5–8; Figs. 1 and 2). By contrast, sites were rather well interspersed across the basin in terms of sea-based and total cumulative human impact scores (Fig. 2). Differences among sites other than those due to local human influences (e.g., climate, oceanography, latitude) were, however, accounted for by the statistical analyses (see below).

The effects of *C. cylindracea* on the percentage cover of morphological groups and on the total cover and species richness of native assemblages were analyzed, separately for extant and recovering assemblages, by means of linear mixed-effect models (*Zuur et al., 2009*). The experimental removal of the invader (*C. cylindracea* removed vs *C. cylindracea* present), cumulative sea-based and land-based human impact scores and their interaction with the *C. cylindracea* treatment were treated as fixed factors. The site was included in the analyses as a random factor, in order to formally assess any potential bias due to unmeasured factors (not related to human impacts) varying among sites. Assumptions of linearity and homogeneity of variances were checked through inspection of plots of residuals vs fitted values and quantile–quantile plots. Log transformation of cover values was effective in improving linearity.

Canopy-forming species, belonging to the genera *Cystoseira* and *Sargassum*, were present only at three locations (i.e. 1, 2 and 4), preventing a test of variations in the effects of *C. cylindracea* on these algal forms along human impact gradients. The effects of *C. cylindracea* on canopy-formers were therefore simply tested by means of a 3-way ANOVA, including the factors *C. cylindracea* treatment (present versus removed; fixed), assemblage (extant versus cleared; fixed) and site (random, with 3 levels). All analyses were done in R 3.2.3 (*R Development Core Team, 2013*) using the libraries lme4, lmerTest and GAD.

## RESULTS

The removal of *C. cylindracea* had no effect on the cover of encrusting algae, turf-forming algae and sessile invertebrates (Table 1A), nor on the total cover and species
**Table 1 Linear-mixed models assessing the effects of *C. cylindracea*, land-based and sea-based cumulative human impacts on extant assemblages.** (A) percentage cover of encrusting algae, turfs and sessile invertebrates; (B) total cover and species richness. Coefficients, Standard Errors (*SE*) and *p*-values are reported for fixed effects, while variance ($\sigma^2$) and Standard Deviation (*SD*) are reported for random effects. Analyses of encrusting algae, sessile invertebrates and total cover are on log-transformed data.

**A) Morphological groups**

|  | Encrusting algae | | | Algal turfs | | | Sessile invertebrates | | |
|---|---|---|---|---|---|---|---|---|---|
| *Fixed effects* | Estimate | *SE* | *P* | Estimate | *SE* | *P* | Estimate | *SE* | *P* |
| Intercept | 1.470 | 0.994 | 0.165 | 57.49 | 60.37 | 0.365 | 1.252 | 1.760 | 0.488 |
| −*Caulerpa* = −C | 0.464 | 0.876 | 0.598 | 44.24 | 32.35 | 0.174 | −1.298 | 1.801 | 0.473 |
| Land-based score = L | −0.219 | 0.076 | **0.014** | 5.39 | 4.62 | 0.272 | −0.192 | 0.135 | 0.174 |
| Sea-based score = S | −0.017 | 0.225 | 0.942 | −7.51 | 13.69 | 0.596 | −0.067 | 0.399 | 0.868 |
| −C. × L | 0.049 | 0.067 | 0.471 | 2.27 | 2.47 | 0.361 | −0.255 | 0.138 | 0.068 |
| −C. × S | −0.109 | 0.199 | 0.585 | −9.06 | 7.32 | 0.220 | 0.569 | 0.409 | 0.168 |
| *Random effects* | $\sigma^2$ | *SD* | | $\sigma^2$ | *SD* | | $\sigma^2$ | *SD* | |
| Site | 0.044 | 0.209 | | 226.1 | 15.04 | | 0.106 | 0.326 | |
| Residual | 0.139 | 0.373 | | 188.2 | 13.72 | | 0.587 | 0.766 | |

**B) Community response variables**

|  | Total cover | | | Species richness | | |
|---|---|---|---|---|---|---|
| *Fixed effects* | Estimate | *SE* | *P* | Estimate | *SE* | *P* |
| Intercept | 2.545 | 0.178 | **0.000** | 22.422 | 6.405 | **0.003** |
| −*Caulerpa* = −C | −0.146 | 0.170 | 0.392 | 0.395 | 6.815 | 0.954 |
| Land-based score = L | −0.022 | 0.014 | 0.137 | −2.333 | 0.490 | **0.000** |
| Sea-based score = S | −0.124 | 0.040 | **0.009** | −1.467 | 1.453 | 0.328 |
| −C. × L | −0.008 | 0.013 | 0.540 | −0.249 | 0.521 | 0.634 |
| −C. × S | 0.057 | 0.038 | 0.143 | 0.403 | 1.546 | 0.795 |
| *Random effects* | $\sigma^2$ | *SD* | | $\sigma^2$ | *SD* | |
| Site | 0.001 | 0.036 | | 1.288 | 1.135 | |
| Residual | 0.005 | 0.072 | | 8.404 | 2.899 | |

richness of extant benthic assemblages (Table 1B). The cover of encrusting algae and species richness decreased at increasing levels of land-based impacts (Tables 1A and 1B; Figs. 3A and 3B) whilst the total cover of extant assemblages declined significantly with increasing levels of human sea-based impacts (Table 1B; Fig. 3C).

The effects of the removal of *C. cylindracea* on the cover of encrusting macroalgae in recovering plots shifted from positive at the most pristine sites to negative at those more impacted by land-based human activities (Table 2A; Fig. 4A). By contrast, the effects of the removal of *C. cylindracea* on algal turf cover were negative at the site least impacted by land-based activities, but became increasingly positive towards the upper end of the human disturbance gradient investigated (Table 2A; Fig. 4B). The cover of sessile

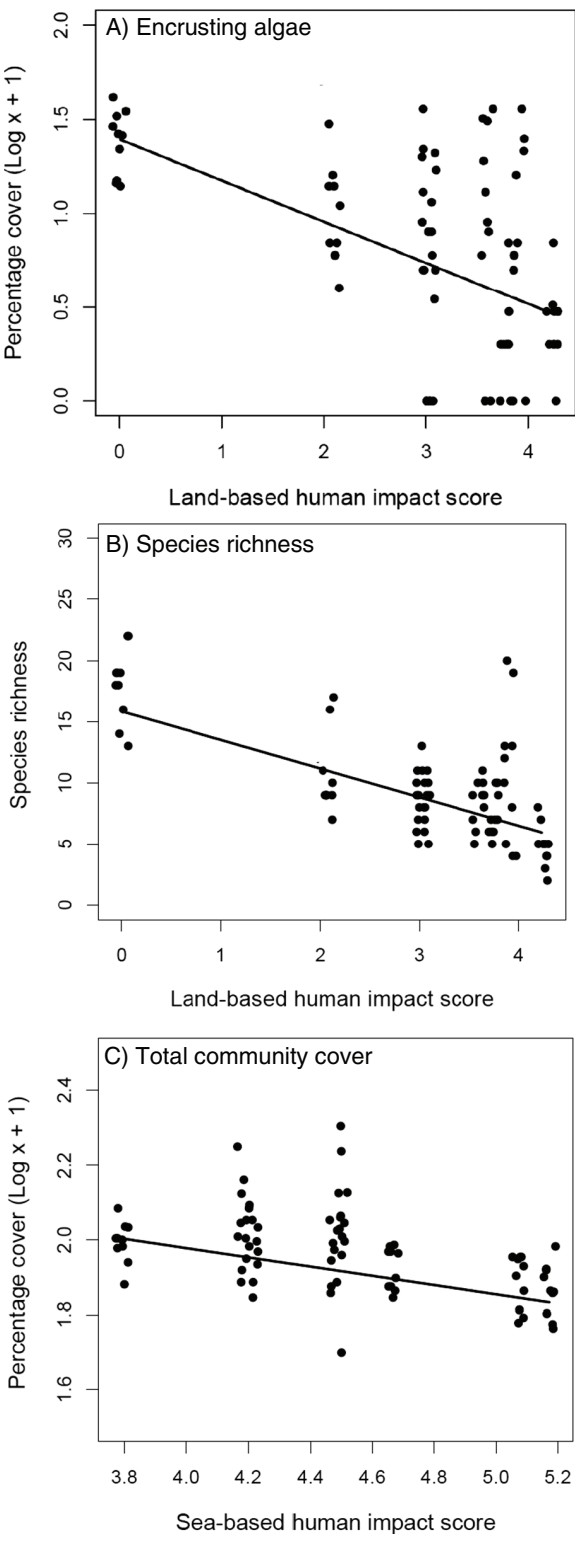

**Figure 3 Extant assemblages.** Relationship between land-based cumulative human impact score and (A) cover of encrusting algae (log scale) and (B) species richness; (C) relationship between sea-based cumulative human impact score and total community cover (log scale). Jittering is used to avoid overplotting.

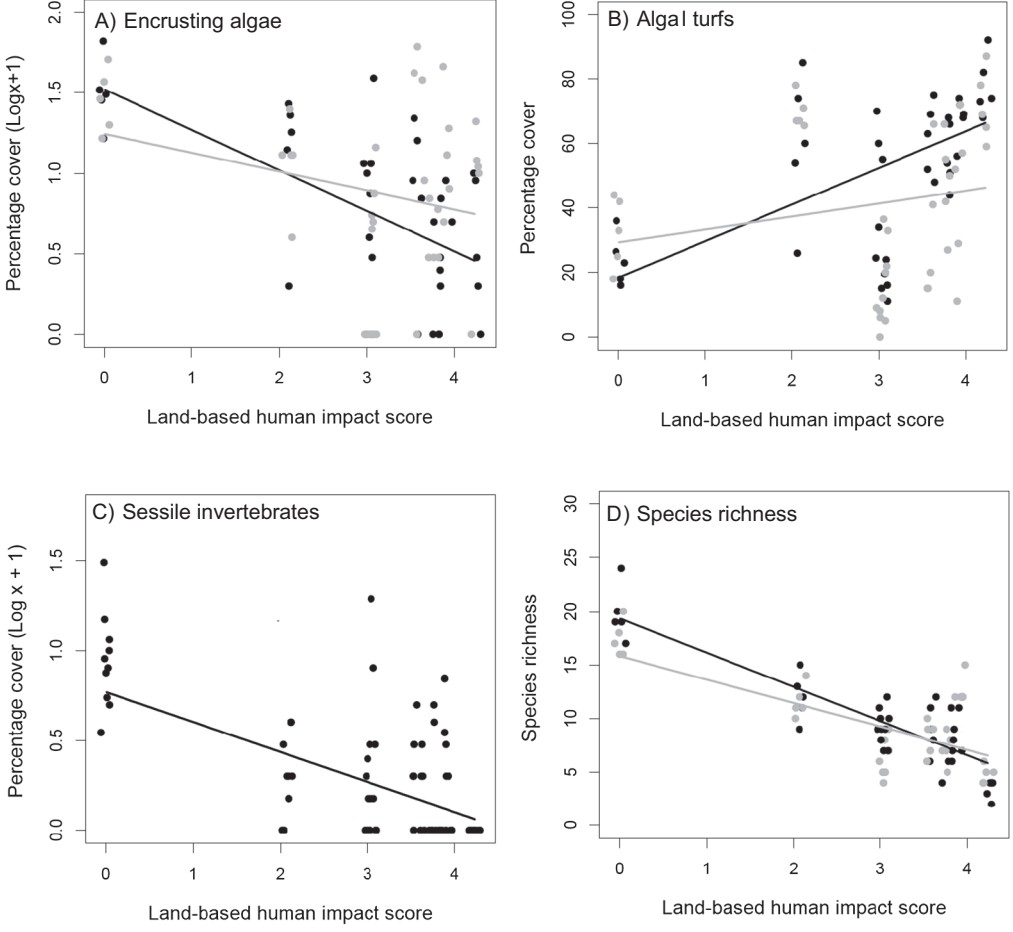

**Figure 4 Recovering assemblages.** Relationship between land-based cumulative human impact score and (A) cover of encrusting algae, (B) cover of algal turfs and (D) species richness, in plots from which *C. cylindracea* was removed (black circles) or left untouched (grey circles). (C) relationship between land-based cumulative human impact score and cover of sessile invertebrates. Jittering is used to avoid overplotting.

invertebrates in recovering plots was not affected by the removal of *C. cylindracea* but decreased with increasing levels of land-based impacts (Table 2A; Fig. 4C). Finally, the cover of canopy-forming macroalgae was greater in plots from which *C. cylindracea* was removed (mean ± SE = 18.9 ± 4.41) than in those in which it was left untouched (mean ± SE = 9.68 ± 2.83), consistently among study sites and between control and cleared plots (Table S2).

The total cover of recovering assemblages was not affected by the removal of *C. cylindracea* and did not show any significant relationship with either land-based or sea-based human impacts (Table 2B). By contrast, the effects of removing *C. cylindracea* on species richness in cleared plots varied according to the intensity of land-based human impacts (Table 2B). The removal of *C. cylindracea* caused a marked increase in species richness at relatively pristine sites, with a trend for this positive effect to weaken when moving towards more impacted areas and, indeed, to shift to negative

**Table 2 Linear-mixed models assessing the effects of *C. cylindracea*, land-based and sea-based cumulative human impacts on recovering assemblages.** (A) percentage cover of encrusting algae, turfs and sessile invertebrates; (B) total cover and species richness. Coefficients, standard errors (*SE*) and *p*-values are reported for fixed effects, while variance ($\sigma^2$) and standard deviation (*SD*) are reported for random effects. Analyses of encrusting algae, sessile invertebrates and total cover are on log-transformed data.

**A) Morphological groups**

| | Encrusting algae | | | Algal turfs | | | Sessile invertebrates | | |
|---|---|---|---|---|---|---|---|---|---|
| *Fixed effects* | Estimate | *SE* | *P* | Estimate | *SE* | *P* | Estimate | *SE* | *P* |
| Intercept | 1.972 | 1.234 | 0.140 | 91.61 | 57.59 | 0.143 | 0.805 | 0.597 | 0.201 |
| −*C. cylindracea* = −C | −0.657 | 0.867 | 0.451 | 2.23 | 36.37 | 0.951 | −0.649 | 0.567 | 0.256 |
| Land-based score = L | −0.118 | 0.094 | 0.239 | 4.01 | 4.41 | 0.385 | −0.167 | 0.046 | **0.003** |
| Sea-based score = S | −0.161 | 0.280 | 0.578 | −13.86 | 13.06 | 0.314 | −0.008 | 0.136 | 0.956 |
| −C. × L | −0.136 | 0.066 | **0.044** | 7.32 | 2.78 | **0.010** | −0.060 | 0.043 | 0.171 |
| −C. × S | 0.209 | 0.197 | 0.291 | −2.93 | 8.25 | 0.724 | 0.191 | 0.129 | 0.142 |
| *Random effects* | $\sigma^2$ | *SD* | | $\sigma^2$ | *SD* | | $\sigma^2$ | *SD* | |
| Site | 0.083 | 0.288 | | 192.2 | 13.86 | | 0.014 | 0.119 | |
| Residual | 0.136 | 0.369 | | 239.4 | 15.47 | | 0.058 | 0.241 | |

**B) Community response variables**

| | Total cover | | | Species richness | | |
|---|---|---|---|---|---|---|
| *Fixed effects* | Estimate | *SE* | *P* | Estimate | *SE* | *P* |
| Intercept | 2.359 | 0.277 | **0.000** | 28.058 | 6.813 | **0.002** |
| −*C. cylindracea* = −C | −0.108 | 0.245 | 0.660 | −3.329 | 4.330 | 0.444 |
| Land-based score = L | −0.029 | 0.021 | 0.196 | −2.187 | 0.521 | **0.002** |
| Sea-based score = S | −0.081 | 0.063 | 0.221 | −2.723 | 1.545 | 0.109 |
| −C. × L | 0.014 | 0.019 | 0.460 | −1.000 | 0.331 | **0.004** |
| −C. × S | 0.029 | 0.056 | 0.604 | 1.521 | 0.982 | 0.126 |
| *Random effects* | $\sigma^2$ | *SD* | | $\sigma^2$ | *SD* | |
| Site | 0.003 | 0.058 | | 2.681 | 1.637 | |
| Residual | 0.011 | 0.104 | | 3.392 | 1.842 | |

towards the upper end of the impact gradient analyzed (Fig. 4D). In particular, removing *C. cylindracea* at the most pristine site caused an increase in species richness of about 14%. In order to assess whether this pattern was driven by the particularly high species richness in one single *C. cylindracea* removal plot (species richness = 24) at the least impacted study site (Fig. 4D), the analysis was repeated after the exclusion of this potential outlier. The interaction between *C. cylindracea* treatment of land-based human impact was also significant in this analysis (Table S1).

## DISCUSSION

The seaweed, *Caulerpa cylindracea*, with a distribution spanning ~10° of latitude and more than ~30° of longitude, is one of the most successful invaders in the Mediterranean
(*Verlaque et al., 2004*; *Piazzi et al., 2005*; *Bulleri et al., 2011*). The removal of this seaweed from extant assemblages had positive effects on canopy-forming species at sites little exposed to human activities, where these algal forms were found (i.e. sites 1, 2 and 4). Canopy stands can provide habitat for a variety of understorey species, including sessile invertebrates and encrusting species, which can be rapidly swamped by fast-growing algal turfs in full light conditions (*Benedetti-Cecchi et al., 2001*; *Bulleri et al., 2002*). Here, the increase in the cover of canopy-forming macroalgae elicited by the removal of *C. cylindracea* may have been too small to foster the recovery of understorey species and, hence, to influence community properties, such as total cover or species richness.

Invasive species can trigger regime shifts, thus generating changes that can be difficult to reverse simply through the removal of the invader (*Gaertner et al., 2014*). Previous experimental work has indeed shown that the removal of *C. cylindracea* alone was little effective in promoting the recovery of architecturally complex macroalgal species (*Bulleri et al., 2010*). Here, the removal of *C. cylindracea* and that of the assemblages that developed under its dominance, likely reinstating more favorable environmental conditions, caused marked changes to native assemblages which varied according to the severity of land-based human impacts.

At pristine sites, preventing re-invasion of cleared plots by *C. cylindracea* favoured the recovery of canopy-forming and encrusting macroalgae, whilst it depressed that of algal turfs, ultimately increasing species richness. *C. cylindracea* is likely to enhance the competitive ability of algal turfs in respect to other components of native assemblages, such as canopy-forming and encrusting macroalgae, through the alteration of abiotic conditions (e.g. enhanced trapping of sediments; *Piazzi et al., 2007*). Our results suggest that altered abiotic conditions persisted, however, beyond the removal of the invader and significant changes at the community level emerged only after legacy effects of invasion were disrupted by an intense disturbance event (i.e. clearing of extant assemblages).

In contrast, at more impacted sites, algal turfs responded positively to the removal of *C. cylindracea*. Thus, the effects of *C. cylindracea* on algal turfs shifted from positive at pristine sites to negative at degraded sites. Turfs are, in fact, generally dominant along urbanized temperate coasts worldwide, where stronger competitors have declined as a consequence of human alteration of abiotic conditions (*Gorman, Russell & Connell, 2009*; *Bulleri et al., 2016*). Under these circumstances, algal turfs do not rely on *C. cylindracea* to reduce competition pressure from native species and competitive effects from the invader prevail.

Seminal work by *Connell (1961)* has shown that community regulation by competitive interactions is more likely under relatively benign environmental conditions. A diminished role of competition at increasing levels of environmental stress, as postulated by the Stress Gradient Hypothesis (*Bertness & Callaway, 1994*), has been documented across a variety of terrestrial (*Grime, 1974*; *Huckle, Marrs & Potter, 2002*; *Brooker et al., 2008*) and marine systems (*Bulleri, 2009*). In this light, the presence of *C. cylindracea* would reduce the number of species in native communities which assembly is mostly

structured by competition, whereas it would have little effect on urban or extra-urban reefs, where filtering by adverse physical conditions limits native species richness (e.g. dominance by filamentous algal turfs; *Tamburello et al., 2012*; *Tamburello et al., 2015*).

Alternatively, the weak response of assemblages to the removal of *C. cylindracea* and to the provision of free space (through the clearing of plots) at more impacted sites could be due to the unavailability of propagules of potential native colonizers other than those already well established locally. Very low species diversity at these sites suggests, in fact, that local assemblages were composed of a few tolerant turf-forming species. Low propagule availability of native species has been shown to hamper restoration both in terrestrial and aquatic environments (*Hansson et al., 1998*; *Prober et al., 2009*). By contrast, at more pristine sites, the removal of *C. cylindracea*, in combination with the provision of free space, may have created the opportunity for rarer species within the local pool to expand their distribution.

The cover of *C. cylindracea* largely varied among study sites, without any apparent relationship with the intensity of either land- or sea-based human impact scores. Removing *C. cylindracea* generated more positive effects at pristine sites (e.g. Capraia and Elba Islands) despite its relatively low cover. By contrast, it had weak effects at more impacted sites, where its cover in re-invaded cleared plots varied by one order of magnitude (between ~80% at Vrsar and ~4% at Quercianella). The effects of *C. cylindracea* on native species richness were, thus, largely independent from its abundance. Previous studies have found that low cover of this seaweed can cause alterations in native macroalgal assemblages (*Bulleri et al., 2010*). Our findings suggest that features of native assemblages (e.g. species composition and relative abundance) were more important than the abundance of *C. cylindracea* in determining the extent of its effects on the native biota. There is mounting evidence that the effects of aquatic invasive species on the native biota may be density-dependent (*Thomsen et al., 2011*; *Gribben et al., 2013*). Density-dependence in the effects of invasive seaweeds might be, however, community-specific. Being modulated by the features of the recipient community, the intensity of the effects of an invader may not scale with its abundance consistently across communities that largely vary in species identity and relative abundance.

Irrespective of the presence/absence of *C. cylindracea*, human impact scores were good predictors of changes in total cover and species richness of extant and recovering assemblages. In particular, the total cover declined along the sea-based human impact score, whilst the cover of encrusting algae and sessile invertebrate and species richness declined along the land-based human impact score, suggesting that different human activities may influence different features of natural communities. Sea-based activities, such as fisheries and shipping activities seem to limit community productivity; mechanical damage due to trawling, operation of gillnets or anchoring, associated with degradation of environmental conditions due to pulse perturbations such as oil spills and ballast water release, may reduce the overall productivity of benthic assemblages, without causing major alterations in species richness. Land-based human impacts, such as inorganic pollution, water turbidity, enhanced nutrient loading and sedimentation would, by contrast, reduce the abundance and number of sessile

species able to thrive on shallow rocky reefs (*Benedetti-Cecchi et al., 2001*; *Airoldi & Beck, 2007*; *Gorman, Russell & Connell, 2009*). Decreased species richness is not necessarily associated with decreases in total cover, as compensation mechanisms among species characterized by different susceptibility to environmental stress can sustain the standing biomass (*Ernest & Brown, 2001*).

In summary, our study experimentally supports the predictions from the meta-analysis of *Tamburello et al. (2015)* and suggests that the effects of invasive seaweeds are likely to be more negative at relatively well preserved sites. In addition, it suggests that land-based human impacts might be a better predictor of invaders' effects compared to sea-based impacts. Encompassing a wider human impact gradient and assessing variation among sites assigned a similar impact score will be crucial to gain conclusive evidence of the sign and strength of the relationship between the severity of human impacts and the effects of invasive seaweeds on the native biota. Nonetheless, following a precautionary principle, implementing the control of those activities (e.g. recreational boating, aquaculture) that can favor the spread of invasive seaweeds into relatively pristine environments could be considered strategic to preserve the biodiversity of temperate shallow rocky reefs.

## ACKNOWLEDGEMENTS

We thank G. Bellistri, C. Ravaglioli, G. Ghedini and L. Tamburello for help with fieldwork and C. Ravaglioli, M. Dal Bello and two anonymous reviewers for commenting on earlier ms drafts. We also thank Rosaria Rizza (Director of Consorzio Plemmirio MPA) and her staff (E. Di Pietro, G. Mazza, M. Moschella e L. Pasolli) for helpful assistance provided during the study.

### Funding

This work has been supported by funds from the EU project VECTORS (Vectors of Change in Oceans and Seas Marine Life, Impact on Economic Sectors) (FP7/2007–2013) and the Ministry of Science, Education and Sport of the Republic of Croatia (Project 098-0982705-2732). The funders had no role in study design, data collection and analysis, decision to publish, or preparation of the manuscript.

### Grant Disclosures

The following grant information was disclosed by the authors:
EU project VECTORS (Vectors of Change in Oceans and Seas Marine Life, Impact on Economic Sectors): FP7/2007–2013.
Ministry of Science, Education and Sport of the Republic of Croatia: 098-0982705-2732.

### Competing Interests

The authors declare that they have no competing interests.

## Author Contributions

- Fabio Bulleri conceived and designed the experiments, performed the experiments, analyzed the data, contributed reagents/materials/analysis tools, wrote the paper, prepared figures and/or tables.
- Fabio Badalamenti performed the experiments, contributed reagents/materials/analysis tools, reviewed drafts of the paper.
- Ljiljana Iveša performed the experiments, contributed reagents/materials/analysis tools, reviewed drafts of the paper.
- Barbara Mikac performed the experiments, reviewed drafts of the paper.
- Luigi Musco performed the experiments, reviewed drafts of the paper.
- Andrej Jaklin performed the experiments, reviewed drafts of the paper.
- Alex Rattray analyzed the data, reviewed drafts of the paper.
- Tomás Vega Fernández performed the experiments, reviewed drafts of the paper.
- Lisandro Benedetti-Cecchi contributed reagents/materials/analysis tools, reviewed drafts of the paper.

## Field Study Permissions

The following information was supplied relating to field study approvals (i.e., approving body and any reference numbers):

Fieldwork along the Italian coast was conducted with permits from Comune di Livorno (Prot. N° 71719), Ente Parco Arcipelago Toscano (Prot. N. 1797) and Area Marina Protetta Plemmirio (CNR-IAMC-Consorzio Plemmirio convention N° 0007139). Fieldwork along the Croatian coast was conducted with permits from the Ministry of Culture (UP/I-612-07/11-33/1015; 532-08-01-01/1-11-04; from 25 October 2011).

## Data Deposition

The raw data has been supplied as a Supplemental Dataset.

## Supplemental Information

Supplemental information for this article can be found online at http://dx.doi.org/10.7717/peerj.1795#supplemental-information.

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
