# Peer review of "The effects of an invasive seaweed on native communities vary along a gradient of land-based human impacts"

_PeerJ, doi:10.7717/peerj.1795_

## Round 0.1 · original submission · Major Revisions

I now have comments back from two referees, both of whom are enthusiastic about your manuscript, but both also point out a couple issues that need to be addressed prior to publication. Both referees agree that only focusing on richness & total seaweed abundance is an issue that needs to be addressed or at least clearly justified in the manuscript. A second issue that the impacted sites are all geographically clustered and adjacent to one another, such that the second reviewer views this as a pseudo-replication with an N of 1. I can see their point, and expect that other readers are likely to have the same response. Thus, I agree that it is important to address this issue directly and that some mention of the limitations of the design be mentioned in the manuscript. That referee also requested to see the revision prior to publication to determine how you have responded to the pseudo-replication issue. For that reason, I am listing this as a major revision that will be returned to the reviewers, although the revisions requested by the referees are not extensive and should be relatively minor in terms of the effort.

Reviewer 1 ·

Basic reporting

The manuscript follows the basic guidelines of PeerJ manuscripts. It is a neat experimental test of a theoretical prediction that emerged from a recent meta-analysis, and it makes clever use of published geo-referenced data on global human impacts.

However, the introduction needs a bit more work to properly place the hypotheses into the broader context of invasion ecology. Specifically, I’d like information on what is known about how C cylindracea impacts Mediterranean rocky reef communities and how you might expect its presence to affect cover and richness. You cover some of this in the Discussion, but please move it to the Introduction to provide a better context for your predictions about how C cylindracea removal will impact benthic cover/richness.

The writing needs tightening in several places where the phrasing is a little confusing. Specifically:
- Abstract, lines 12-17: these two sentences summarizing the results are very confusing. Findings are interesting but really difficult to understand from the way these sentences are written.
- line 40-43, sentence beginning “The impact score…” is quite convoluted.
- line 59: “we predicted the effects…would be negative in relatively pristine areas *to become* neutral or positive…” - stating your hypothesis should be VERY clear and well-written. Perhaps “to become” was meant to read “and become”?

Experimental design

With the exception of the few issues addressed below, the experimental approach is clearly explained and directly addresses the hypotheses.

It is difficult to understand exactly what the treatments were based on their names in the Methods section. Disturbance (control vs removal) and Manipulation (present vs removed) sound similar and redundant. I suggest either 1) something like “reinvasion prevented” vs “reinvasion allowed” instead of “present” and “removal”, which sounds the same as “control” vs “removal”, or 2) rename “control vs. removal” to “control vs. cleared”. Whichever you choose, make sure terminology matches Fig 2 caption and Results.

Please provide some justification for why you used total cover and richness as response metrics. Are those metrics of ecosystem health or “pristineness”? Further, total cover isn’t a great proxy for biomass - what about encrusting or filamentous algae versus large macro algae? This point (Line 54) seems unnecessary. Cover in and of itself is a fine metric. If you wanted biomass data, report biomass data.

“Native cover” is used synonymously with cover of all other invertebrate and algae that are not Caulerpa. Is this true? If so, please state clearly that all other taxa are native.

Validity of the findings

Results are directly linked to the experiments and clearly explained in support/refutation of hypotheses. However, some details of the data are missing and should be included to fully report your findings:
- Instead of selectively showing only figures with significant results, please present all figures of land-based and sea-based scores vs. richness cover for both extant and developing assemblages? This could be easily accomplished in one or two panel figures.
- Please report the actual p-values (Table 1) and not the unnecessarily old-school strategy of only reporting whether they were greater than 0.05 (or less than other cutoffs).

Additional comments

Very interesting study with a clever approach to test land- vs. sea-based human impacts on how invasive species affect native communities. I have only minor quibbles about how the hypotheses is framed and how the data are presented, but these should not preclude publication after revisions.

Reviewer 2 ·

Basic reporting

No comments

Experimental design

No comments

Validity of the findings

No comments

Additional comments

Review Bulleri et al; The effect of an exotic seaweed on native community diversity vary along a gradient of land-based human impacts by Bulleri et al.
Bulleri et al carry out a 15 month removal experiment to test if effects from the invasive seaweed Caulerpa racemosa vary predictable along human impact gradients. They found, in support of results from an earlier meta-analysis, that effects on developing communities (but not established communities) were more negative in relatively pristine areas compared to areas strongly affected by human activities.
The paper is testing an interesting hypothesis and is well-written.
I recommend publication after minor revisions. Below follow first two general comments and then my minor comments follows in chronological order thereafter.
Two major comments
1. Why is the analyses only focusing on richness & total seaweed abundance? These two metrics are rough and insensitive to detect impacts of invaders and human activities. A hypothetical (yet not entirely unrealistic) example; 100% cover of turf forming seaweeds can replace 100% cover of canopy formers along a strong nutrient gradient – resulting in no net changes to richness (1 vs. 1 species) or total abundance of seaweed (100% vs. 100%) – although there clearly are dramatic effect of nutrients on specific species, forms and community structures. In short - you would be much more likely to find strong effects of the invader and human activities if you analyze abundances of specific species, form-functional groups or on multivariate community metrics. Why do you not include analysis of more groups, species and communities – that are more sensitive to detect impacts? Non-significant results are equally important to report of these groups to support the non-significant results already shown for the two coarse metrics.
2. Your test about human impact is ‘pseudo-replicated’ because all the impacted sites (S5-S8 on fig. 1) are located adjacent and very close to each other (i.e. in classical experimental sampling they would be considered 1 not 4 replicates). In an ideal world the 4 impacted sites would be interspersed between the 4 ‘pristine’ sites. Any unmeasured co-varying factor (not related to human impacts) can therefore equally explain your results. For example, your results could instead be explained by hypothetical unmeasured factors, such as biogeographical anomalies, temperature anomalies, distinctive current/wave regimes, natural river runoff, different bedrocks etc. etc. (use your imagination). You should be upfront about this design problem and mentioned it in the method section and then discuss interpretation implications in the discussion (instead of letting it be up to the critical reader to find out her/him-selves)
Minor comments
Title; ‘exotic’ – other places you use non-native and introduced species – avoid extra jargon and just one use one of the terms (just do a search and replace)
L19 ‘assemblage’ – other places you use ‘community’ – avoid extra jargon and just one use one of the terms
Introduction generally. I miss a paragraph outlining the rationale for the test of impacts on both extant and colonizing resident communities (as was the main emphasis in your 2010 Ecology paper).
L28. ‘native’ – other places you use ‘resident’ – avoid extra jargon and just one use one of the terms
L30. ‘…direction and magnitude of impact… from quantitative synthesis of the literature… ‘ The first two references (S&H 07 and W&S 07) do not do this – these two studies used classic vote-counting – which may give a direction but certainly not a magnitude.
L44. Should be ‘enhanced sedimentation’
L49. You write …’in comparison to seabased impacts’ – but there are no previous lines that set you up for this statement (i.e. you need 1-2 lines first to describe seabased impacts – to understand the following sentences that focus on landbased impacts)
L102. Can you explain the rationale for doing log x+1 transformation of human impacts? (standardization is obvious – by why this strong transformation). This transformation downgrade strong human impacts – potentially making it more difficult to detect human impacts - Why do you (and Micheli 13) do this harsh transformation?
L104 ‘Worth stressing’ – could be worded better
Fig. 3 and 4. I suggest to use different symbols too (in addition to different colors). It does not show on B/W prints.

---

## Round 0.2 · accepted · Accept

Thank you for your careful revision of the submission. I find myself in agreement with the referee who requested to see the revision - the revised manuscript is an even stronger contribution and is now ready for acceptance. The referee has a couple of suggestions that you may wish to incorporate into the final manuscript, but they are minor enough that they could likely be incorporated at the proof stage. I leave it to you if you wish to incorporate these suggested revisions prior to moving into production.

Reviewer 2 ·

Basic reporting

Review ‘The effects of an invasive seaweed on native communities….’ By Bulleri et al
I am happy with the revised manuscript– indeed, the paper is even stronger now that they show divergent effects of invaders on encrusting alga and turf (compared with just showing total cover in first version).
I just found a few things to correct
L101-103. You state a specific prediction about comparing impact strength on extant vs. cleared communities. However, in the method section (l161) you state you analyse these two parts of the experiment separately (i.e. you don’t show statistical results to back up your prediction). I would either remove the prediction from intro, add statistical comparison (or last option – say something here that this last prediction is only analysed by comparing graphs)
L212. Is one OF the
L225-228. Complex sentence – can you simplify – or make into two?
L235. I think the ‘can’ in the end of the line should be deleted?
L246-249. You state that the SGH is documented in terrestrial systems and add supporting references. However, I would argue that its even better documented in marine systems – with all of Bertness original work in saltmarshes and rocky intertidal systems
L298 change ‘in respect’ to ‘compared’

Experimental design

See my comments to original manuscript - the revised manus is fine

Validity of the findings

See my comments to original manuscript - the revised manus is fine

Additional comments

See my comments to original manuscript - the revised manus is fine